# Language Grounding with 3D Objects

**Jesse Thomason**[*]
University of Southern California

**Mohit Shridhar**[*]
University of Washington

**Yonatan Bisk**
Carnegie Mellon University

**Chris Paxton**
NVIDIA

**Luke Zettlemoyer**
University of Washington

**Abstract:**

Seemingly simple natural language requests to a robot are generally underspecified, for example *Can you bring me the wireless mouse?* Flat images of candidate mice may not provide the discriminative information needed for *wireless*. The world, and objects in it, are not flat images but complex 3D shapes. If a human requests an object based on any of its basic properties, such as color, shape, or texture, robots should perform the necessary exploration to accomplish the task. In particular, while substantial effort and progress has been made on understanding explicitly visual attributes like color and category, comparatively little progress has been made on understanding language about shapes and contours. In this work, we introduce a novel reasoning task that targets both visual and non-visual language about 3D objects. Our new benchmark **S**hape**N**et **A**nnotated with **R**eferring **E**xpressions (SNARE) requires a model to choose which of two objects is being referenced by a natural language description.[2] We introduce several CLIP-based [1] models for distinguishing objects and demonstrate that while recent advances in jointly modeling vision and language are useful for robotic language understanding, it is still the case that these image-based models are weaker at understanding the 3D nature of objects – properties which play a key role in manipulation. We find that adding view estimation to language grounding models improves accuracy on both SNARE and when identifying objects referred to in language on a robot platform, but note that a large gap remains between these models and human performance.

**Keywords:** Benchmark, Language Grounding, Vision, 3D

## 1 Introduction

Joint language and vision models are often trained on image captions which have a bias towards canonically "visual" attributes of objects, such as color, rather than functional ones like shape. Image captions omit properties that require understanding objects as 3D, rather than flat, concepts. In this work, we show how the effect of this disconnect is that while robotics research has benefited greatly from advances in computer vision, vision-and-language models cannot always be directly applied to robotics. People use geometric and physical properties of objects when describing them. For example, a robot carrying out a request to *Bring me the mug with the wide handle* needs to be able to spot the *wide handle*, even if the mug rests on a countertop with the handle oriented out of view, but a robot can rotate such an object to investigate in further detail. Generally, identifying objects based on natural language is more challenging from non-canonical viewpoints, which are common for robots in home, office, and industrial environments.

Figure 1: Deciding between 3D objects described in natural language involves rotating objects to see multiple faces.

---

[*]Equal contributions.

[2] https://github.com/snaredataset/snare

5th Conference on Robot Learning (CoRL 2021), London, UK.

The codification and commodification of ResNet [2] architectures pretrained on ImageNet [3] has yielded immense progress in off-the-shelf computer vision techniques. Such representations provide a strong foundation for visual reasoning. With the introduction of pretrained Transformer [4] architectures like BERT [5], natural language processing has similar off-the-shelf tools for representing language. Modern visual language grounding is done through massive, internet-scale pretraining to combine these strengths: aligning images with their captions and content descriptions in natural language using Transformer models [6, 1, 7, 8].

However, internet images suffer from a type of *reporting bias*. These images are captured by sighted humans, from human-centric vantage points [9]. There is a domain shift between such canonical depictions of objects in images and objects as seen through a robot camera [10, 11], as studied in computer vision between such images and those captured by the blind [12]. For example, images of *mugs* online almost exclusively feature a full or three-fourths view of the handle, while a mug in the wild can easily be oriented such that the handle is occluded from the camera by the mug body.

A person requesting a mug with a wide handle has a *mental model* of the referent object that includes its full 3D spatial information; humans do not imagine objects only in 2 dimensions. Further, someone trying to retrieve said mug will differentiate it from other mugs on the countertop by viewing the mugs from different angles to inspect the handles.

Our goal is to similarly imbue robots with three dimensional notions of language grounding, by encouraging them to *rotate* an object when selecting the referent of a language description. We introduce the **S**hape**N**et **A**nnotated with **R**eferring **E**xpressions (SNARE) dataset, which provides discriminative natural language descriptions of 3D ShapeNet [13] object models. SNARE includes both *visual* descriptions that focus on colors and object categories and *blindfolded* descriptions that focus on shapes and parts. We finetune large off-the-shelf models to select the correct object given its description, and find that estimating the input views as an auxiliary prediction improves accuracy.

We introduce **La**nguage **G**rounding through **O**bject **R**otation (LAGOR) which performs view estimation as an auxiliary loss to encourage 3D object understanding (Section 4) during the SNARE task. We find that using multiple views to select the correct object improves over single-view object selection, using the large-scale vision-and-language CLIP [1] model as a backbone to score how well an image and language MATCH. We show that models taking in multiple views of objects objects, while performing auxiliary view estimation, can result in accuracy approaching that of models consuming panoramic views of objects, while also being more realistic for a robot.

Our key contributions are:

- SNARE, a benchmark dataset for identifying 3D object models given natural language referring expressions including *visual* and *blindfolded* descriptions;
- Baseline models for SNARE demonstrating both zero-shot and fine-tuned performance of state-of-the-art vision-and-language models; and
- LAGOR, an initial SNARE model that looks at two random views of an object, estimates the angle of those views, then makes a referent prediction, which we demonstrate using a robot.

## 2   Related Work

Using natural language to work with robot partners is a long-standing goal in robotics [14]. We argue that internet-scale, pretrained vision-and-language models offer a powerful starting point for human-robot collaboration. Unlike 2D internet images, physical objects can be picked up and moved by robot agents. Agents can perform information-seeking behaviors on objects to improve their world [15, 16, 17] and language understanding [18, 19, 20]. Such world interaction is inextricable from language grounding [21], motivating language annotations for higher-fidelity referents than static images. We introduce SNARE, comprised of language referring expressions for 3D objects, and the LAGOR model to combine language and multiple object views to achieve better 3D understanding. SNARE extends lines of work tying language to static images, 3D object models, and even physical objects. We demonstrate that LAGOR generalizes to physical objects manipulated by a tabletop robot §5.2. LAGOR is inspired by models that perform information-seeking actions informed by learned world models to better achieve goals.

**Image-based Object Identification** Object classification [3, 2] is the first step towards image captioning and visual question answering [22]. Particular object instances can be found with region seg-

mentation models such as MaskRCNN [23], enabling object referent tasks such as GuessWhat!? [24]. Combining visual recognition with Transformer-based language understanding to jointly attend to language and visual tokens leads to improved downstream performance on many vision-and-language tasks [6, 25, 8]. Joint embedding approaches that learn a shared subspace for representing language and vision tokens [26] also achieve state-of-the-art performance when trained at scale [1]. We show that these large-scale, pretrained models fall short of human performance on the SNARE task, and that synthesizing 2D views from multiple vantage points improves object identification performance.

**Language Grounding in 3D** Prior works have associated single-word attributes with 3D object models based on latent representations of 3D meshes [27]. To learn spatial language, vision-and-language navigation (VLN) [28] models infer navigation actions from instructions and visual observations in 3D simulated worlds. Such tasks can be extended to include simulated world interactions such as picking and placing objects [29] and using appliances and tools [30]. Models for these tasks extend Transformer representations to include action taking [31, 32] and invoke object classification methods to create a semantic understanding of the world [33]. SNARE poses a complementary challenge, providing data for selecting referent objects in the presence of distractors by taking into account the multiple views possible of objects in 3D space.

The work most similar to ours is Shape-Glot [34], a collection of referring expressions for ShapeNet objects to discriminate between two distractors. While Shape-Glot is similar in spirit, it contains training data only for `chair` objects, and tests on four additional categories. By contrast, SNARE spans 262 distinct object categories (Table 1). ShapeGlot focuses on a model's abilities to learn particular parts of objects, such as chair arms and legs, while SNARE aims to ground language more generally to 3D features of objects span-

| Data | Fold | # Cats | # Objs | # Ref Exps |
|------|------|--------|--------|------------|
| ShapeGlot | Chairs | 1 | 4,511 | 78,789 |
| | Other | 4 | 200 | 400 |
| | **Total** | **5** | **4,711** | **79,189** |
| SNARE | Train | 207 | 6,153 | 39,104 |
| | Val | 7 | 371 | 2,304 |
| | Test | 48 | 1,357 | 8,751 |
| | **Total** | **262** | **7,881** | **50,159** |

Table 1: Fold summaries in SNARE, with the Shape-Glot benchmark size for contrast.

ning color, shape, category, and parts. Further, ShapeGlot models take in a pointcloud representation of objects along with visual features, where our LAGOR model does not assume access to the 3D model of an object and instead operates solely on different 2D views achieved through object rotation.

**Physical Object Identification** Modeling the connections between language to physical objects enables robots to identify objects for manipulation by color, shape, and category [35, 36]. The physical forces and sounds objects make during manipulation actions can also be associated with words such as *rattling* and *heavy* for multimodal understanding beyond vision [19, 37]. Prior work has gathered language annotations for the YCB Benchmark object set [38] to explore how language descriptions provide priors on object affordances [39]. In our tabletop robot experiments, we use camera views of novel objects to evaluate zero shot transfer of LAGOR to robot camera views of real objects, which are processed by an off-the-shelf segmentation algorithm.

**Information-Seeking Actions** Embodiment, whether in simulation or the physical world, affords agents the opportunity to seek out new information to help with a given task. By predicting the new information that can be gotten from different actions, agents can *prospect* over potential futures for planning [40, 41, 42]. In the aforementioned VLN task, predicting *what* will be seen along different potential routes facilitates more efficient navigation [43]. Our LAGOR model estimates the angles of object input views, encouraging 3D understanding in the learned object representations.

## 3 ShapeNet Annotated with Referring Expressions (SNARE)

We introduce SNARE, as a new benchmark for grounding natural language referring expressions to distinguish 3D objects. The annotations are collected to complement the ACRONYM[3] [44] grasping dataset and include language that targets both visual and tactile attributes of objects. Our goal is to enable future research to ground both from multi-view vision, as in this work, and directly from robotic grasp contact point data (Section 6).

---

[3]Yes, that dataset title is "ACRONYM" [44].

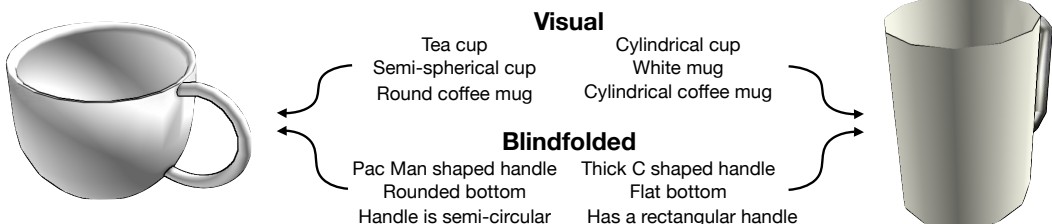

**Visual**

Tea cup · Cylindrical cup
Semi-spherical cup · White mug
Round coffee mug · Cylindrical coffee mug

**Blindfolded**

Pac Man shaped handle · Thick C shaped handle
Rounded bottom · Flat bottom
Handle is semi-circular · Has a rectangular handle

Figure 2: Example object pair and object referring expressions in SNARE. Given a referring expression and one (or more) views of two contrasting objects, a model must decide which object is being referenced by the language. Language annotations are collected in two forms: **Visual** and **Blindfolded**. The latter expressions tasked annotators with describing objects tactilely.

To construct SNARE, we select a subset of 7,897 ACRONYM [44] object models from ShapeNet-Sem [45, 13]. We obtain over 50,000 natural language referring expressions in English for these object models via Amazon Mechanical Turk (AMT).[4] Figure 2 gives an example of referring expressions collected for two `mug` objects used to distinguish between the two.

To elicit referring expressions with high specificity, we frame the annotation as a discriminative task. AMT workers were presented with two object models side-by-side from the same ShapeNet category, for example two different `OutdoorTable` object meshes, and asked to complete sentences like

- The way to tell Object A from Object B is that Object A looks like a(n) ___
- Blindfolded, the way to tell Object A from Object B is that Object B is a(n) ___

with referring expressions. We displayed the object models as GIFs rendered to give a 360 rotating view of each object. The resulting SNARE task asks models to take in one such referring expression and decide whether it applies to Object A or Object B.

Priming in the annotation prompts invokes visual features (*. . . looks like. . .*) and non-visual features (*Blindfolded, . . .*) in the referring expressions. By pairing objects from the same ShapeNet category, referring expressions must go beyond categorical information like *brown dresser* to differentiate one object from the other with higher specificity, similar to ShapeGlot's choice to use distractor objects from a single training category [34]. We collected six referring expressions per object—three primed to be visual and three primed to be blindfolded. Priming for visual expressions mentioned *the name of the object, shapes, and colors*, while for blindfolded mentioned *shapes and parts*. Each referring expression was vetted through a secondary task on AMT where, given the language expression, workers had to correctly select the referent object. Every referring expression in SNARE was correctly associated to its referent object by a majority of such annotators.

SNARE referring expressions average 4.27 words, with blindfolded expressions longer (4.95) than visual expressions (3.63). We estimated that blindfolded expressions use more shape words (14% vs 5% of all words) while visual use more color words (22% vs 1% of all words), by traversing the WordNet [46] hierarchy for each word and noting whether it is a hyponym of *color* or *shape*. This distribution difference is consistent with the priming used to elicit visual and blindfolded expressions.

Each SNARE instance is a tuple of (referring expression, Object A, Object B), and models must select which of Object A or B the referent of the natural language expression. We split these data instances into train, validation, and test folds by ShapeNet category. We ensure that closely related categories such as `2Shelves` and `3Shelves` or `DiningTable` and `AccentTable` are within the same fold. For example, all shelf-related and bed-related categories are sorted into the train and test folds, respectively, so that inter-category information does not leak across folds. More details can be found Section 7.3. Table 1 summarizes the overall data statistics.

## 4 Methods

There has been a recent proliferation of off-the-shelf, increasingly large-scale pretrained vision and language alignment models applicable for language grounding in robotics [1, 6, 7, 8, 47]. We set out to answer two questions about such models, using SNARE as a testbed:

---

[4]Section 7.1 contains additional details about the AMT study.

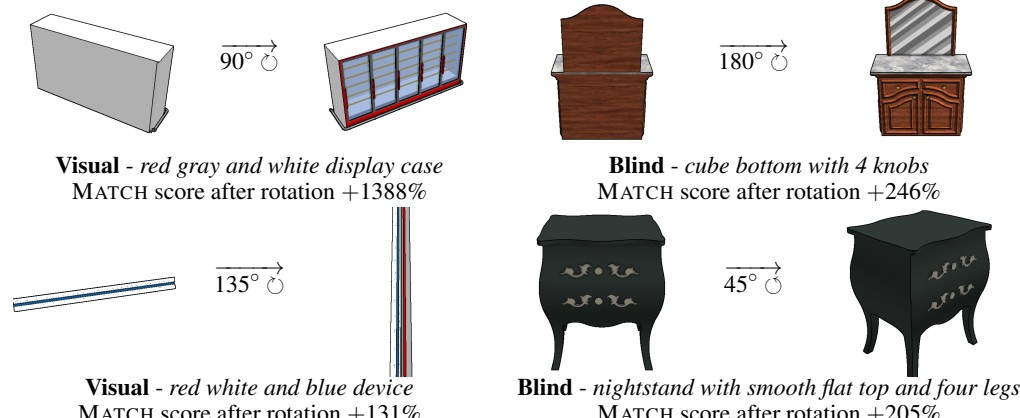

**Visual** - *red gray and white display case*
MATCH score after rotation +1388%

**Blind** - *cube bottom with 4 knobs*
MATCH score after rotation +246%

**Visual** - *red white and blue device*
MATCH score after rotation +131%

**Blind** - *nightstand with smooth flat top and four legs*
MATCH score after rotation +205%

Figure 3: Example MATCH score increases after performing object rotation. Rotations involve exposing colors (top left, bottom left), canonical faces (top right), and parts (bottom right). A robot encountering objects in the wild will find initial object orientations that do not line up with natural language referring expressions for those objects, as reflected in SNARE.

- Can existing language and vision models ground language in the SNARE benchmark?
- Do SNARE models generalize to a robot object selection task better than off-the-shelf models?

To answer the first question, we first train a MATCH module that learns a predictive head on top of a CLIP [1] backbone, and evaluate this module on single and multiple views of 3D objects. To answer the second question, we introduce the LAGOR model, which uses view estimation as an auxiliary loss while predicting language expression referent objects for SNARE, and evaluate on both SNARE §5.1 and a physical robot platform §5.2. LAGOR examines an initial 3D object view and an additional, post-rotation view, striking a balance between single-view models and those that attempt to capture each object's entire 3D structure.

### 4.1 Language-View Match (MATCH) Module

The MATCH module takes in a language expression $L$ and single view of an object, $V_i$, and produces a match score $s(L, V_i)$. To decide whether Object A or Object B is the referent of a language expression on SNARE, we interpret $\text{argmax}_{O \in \{A,B\}} s(L, V_{i,O})$ as the model's prediction.

CLIP [1] serves as a backbone on which we add additional, learnable layers for the SNARE task. We use CLIP's transformer-based sentence encoder to extract language features $\mathbf{l} : \mathbb{R}^{1 \times 512}$. We use CLIP ViT-B/32 to extract visual features for image $V_{i,A}$, $\mathbf{v}_{i,A} : \mathbb{R}^{1 \times 512}$, and image $V_{i,B}$, $\mathbf{v}_{i,B} : \mathbb{R}^{1 \times 512}$. These modality-specific feature vectors are concatenated: $[\mathbf{v}_{i,A}; \mathbf{l}] : \mathbb{R}^{1 \times 1024}$ and $[\mathbf{v}_{i,B}; \mathbf{l}] : \mathbb{R}^{1 \times 1024}$ and independently run through a learnable, multi-layer perceptron that gradually reduces the dimensionality from 1024 to 512, then 256, and finally to a single-dimensional score $s$. This final score comparison training mirrors the multiple-choice formulation used in existing unimodal [5] and multimodal transformers [7]. We keep the CLIP encoders frozen during training.

ShapeNet objects are 3D meshes, incompatible with the 2D input expected by off-the-shelf vision and language models. We sample eight rendered viewpoints around each object at 45 degree increments, and compare off-the-shelf CLIP and MATCH module accuracies when considering single or multiple of these views. To aggregate multiple views, we maxpool over view embedding vectors $\mathbf{v}_{i,A} \ldots \mathbf{v}_{j,A}$.

MATCH is trained with cross-entropy-loss, $\mathscr{L}_s$, to predict a binary label of whether the referring expression matches the object represented in the view image. MATCH is trained for 50 epochs on SNARE, with object views chosen at random during each step.

**Baselines.** We compare our fine-tuned MATCH against a zero-shot CLIP classifier. Instead of fine-tuning CLIP, we use the cosine distance between visual and language features to pick the referred object. That is, we select $\text{argmax}_{O \in \{A,B\}} \mathbf{l} \cdot \mathbf{v}_{i,O}$. We also evaluate against a trained ViLBERT baseline that consumes multiple object views at once, but find that it does not perform as well as the MATCH

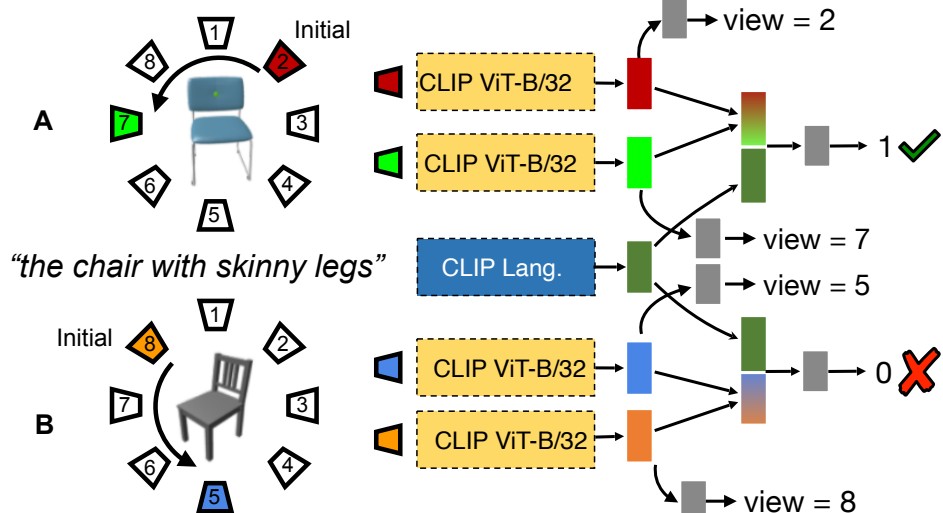

Figure 4: **LaGOR model** relies on the pretrained CLIP [1] architecture as a backbone encoder across multiple views compared to the encoded referring expression. The model sees improvements from auxiliary losses that predict the orientation of the initial and final view.

module; more details about the ViLBERT baseline can be found in Section 7.4. The MATCH model can be thought of as a fine-tuned CLIP model; rather than back-propagating through the CLIP representation itself we learn additional tuning layers on top.

## 4.2 Language Grounding through Object Rotation (LaGOR)

A robot tasked with retrieving an object given a natural language expression should not have to gather a 360 degree view of each candidate referent object to make a decision. We have the intuition that estimating the input viewing angles can enable a model to develop a global reference frame for language grounding against 3D objects (Figure 3). Thus, we propose LaGOR, which takes in two views of each candidate object, performs *view estimation* on those views as an auxiliary prediction (Figure 4), then predicts the language referent.

In addition to predicting the object referent using a pre-trained MATCH module that considers two views, LaGOR predicts the input views of each object using a cross-entropy loss, $\mathscr{L}_v$, against a 1-hot vector representing a discrete set of 8 views at 45 degree offsets. LaGOR learns a multi-layer perceptron that takes in a visual embedding $V_i$ and reduces dimensionality from 512 to 256 to 128 to 64 to a vector of 8 logits representing the 8 discrete object views. The MATCH loss, $\mathscr{L}_s$, is combined with the view estimation loss for a final loss function $\mathscr{L} = \mathscr{L}_v + 0.2 * \mathscr{L}_s$.

## 4.3 Robot Demonstration

We evaluate LaGOR trained on SNARE on a robot platform, taking two random views of objects and evaluating the robot's ability to select which is the correct referent of a language expression (Section 5.2). We compare LaGOR to the off-the-shelf CLIP model that considers a single view of the object only, before performing any view-gathering rotations.

We set two objects on the workspace of a Franka Emika Panda with a wrist-mounted Intel Realsense D414.[5] We capture an initial view of the scene, then segment the objects from the workspace using the Unseen Object Clustering [48] algorithm. We feed the segmented objects as the initial views to LaGOR and CLIP. We move the arm to a second vantage point above the workspace and capture another image of the objects and repeat segmentation to obtain the second views for LaGOR.

---

[5]https://www.franka.de/ and https://www.intelrealsense.com/depth-camera-d415/ respectively

| | | Validation | | | Test | | |
|---|---|---|---|---|---|---|---|
| Model | Views | Visual⊂ | Blind⊂ | All | Visual⊂ | Blind⊂ | All |
| ViLBERT | All | **89.5** | **76.6** | **83.1** | 80.2 | **73.0** | **76.6** |
| CLIP | All | 83.7 ±0.0 | 65.2 ±0.0 | 74.5 ±0.0 | 80.0 ±0.0 | 61.4 ±0.0 | 70.9 ±0.0 |
| MATCH | All | 89.2 ±0.9 | 75.2 ±0.7 | 82.2 ±0.4 | **83.9** ±0.5 | 68.7 ±0.9 | 76.5 ±0.5 |
| CLIP | Single | 79.0 ±0.0 | 63.0 ±0.0 | 71.1 ±0.0 | 74.0 ±0.0 | 59.7 ±0.0 | 67.0 ±0.0 |
| MATCH | Single | **88.4** ±0.4 | **73.3** ±0.6 | **80.9** ±0.4 | 83.2 ±0.3 | 68.0 ±0.5 | **75.8** ±0.3 |
| CLIP | Two | 81.0 ±0.0 | 64.1 ±0.0 | 72.6 ±0.0 | 76.0 ±0.0 | 60.8 ±0.0 | 68.6 ±0.0 |
| MATCH | Two | 89.2 ±0.6 | 74.4 ±0.7 | 81.8 ±0.4 | 83.7 ±0.4 | 68.7 ±0.5 | 76.4 ±0.4 |
| LAGOR | Two | **89.8** ±0.4 | **75.3** ±0.7 | **82.6** ±0.4 | **84.3** ±0.4 | **69.4** ±0.5 | **77.0** ±0.5 |
| Human (U) | All | 94.0 | 90.6 | 92.3 | 93.4 | 88.9 | 91.2 |
| Human (M) | All | 100.0 | 100.0 | 100.0 | 100.0 | 100.0 | 100.0 |

Table 2: **Accuracy on the SNARE benchmark.** Mean accuracy ± standard deviation over 10 seeds. CLIP zero-shot accuracy is well above random chance, and adding more object views improves all models' performance. LAGOR, which needs only two views, making it reasonable to deploy on a robot platform, outperforms two-view MATCH statistically significantly on both the validation and training set, by adding a level of 3D object understanding in the form of view estimation auxiliary losses. Human accuracy is conservatively calculated as the number of SNARE instances for which the correct referent object was identified by *every* voting annotator (**U**nanimous); for all SNARE instances a majority of voting annotators correctly selected the referent (**M**ajority).

## 5 Results

In this section we show that CLIP [1] and ViLBERT [6] select correct 3D object referents in SNARE substantially less often than the MATCH and LAGOR models. Then, we deploy CLIP and LAGOR on a robot tasked with selecting objects conditioned on natural language referring expressions, and find that view estimation and gathering additional object viewpoints improves referent identification.

### 5.1 Performance on SNARE

We train all models for 50 epochs on SNARE, with object views chosen at random during each step, and report the best validation performance across those epochs, as well as the test performance at that best validation epoch checkpoint. We train each model 10 times with a different random seed to estimate performance variance across training runs.

Table 2 gives LAGOR accuracy compared to other two-view alternatives, as well as models taking in all views or single views. We compare to existing models trained (ViLBERT) and zero-shot (CLIP), as recommended in their respective papers. We do not re-run ViLBERT training because it is extremely time-consuming; the same reason we build on CLIP for our model development. To compare models, we perform an unpaired, Welch's two-tailed $t$-test on overall accuracy considering both visual and blindfolded subsets. We perform five such tests in all: for validation and test, LAGOR against two-view MATCH, LAGOR against all-view MATCH, and one additional pooling test (Section 7.5). We perform a conservative Bonferroni multiple-comparison correction applied to threshold of $p < 0.05$.

LAGOR statistically significantly outperforms two-view MATCH on both the validation ($p = 0.0008$) and test ($p = 0.0055$) sets. The modelling difference between LAGOR and MATCH with two views only the view estimation auxiliary losses. LAGOR averages higher accuracy than the MATCH that uses all views on every metric; however, these differences are significant for neither the validation ($p = 0.0844$) or test ($p = 0.0315$) set.[6]

While no models achieve human level accuracy, the performance difference is particularly striking on the blindfolded referring expression subset of data, which lags 5-15% behind visual referring expressions across models. That gap supports our intuition that blindfolded referring expressions capture a complementary and challenging linguistic space currently understudied for vision-language models but key to everyday manipulation. Thereby, SNARE opens interesting avenues for future work exploring the shape and grasp-points of objects.

---

[6]Note that with the Bonferroni correction, significance is reached only with $p < 0.01$ given the desired threshold of 0.05 with 5 tests run.

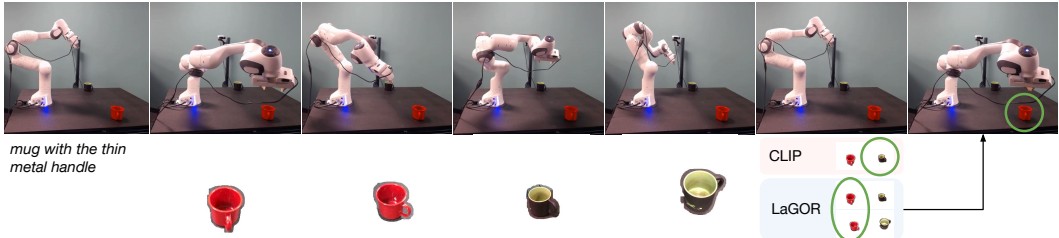

Figure 5: With a physical robot platform, LAGOR outperforms zero-shot CLIP for selecting tabletop referent objects whose views are captured via multiple camera angles and off-the-shelf segmentation.

Notably, zero-shot CLIP performance on SNARE, even when considering only one random view of each object, is well above random chance. While MATCH fine-tuning and LAGOR auxiliary losses improve accuracy over zero-shot CLIP, the zero-shot performance is a good indicator that massively pre-trained language and vision alignment like CLIP could be useful for robotics tasks for which there is little or no in-domain data.

## 5.2 Robot Results

We performed a set of experiments on a real robot (Figure 5). We run single view, zero-shot CLIP against LAGOR on 11 referring expressions and pairs of objects. Two objects at a time were placed on the table, and we selected two arbitrary viewing angles for each object. We used unknown object instance segmentation [48] to segment out each object. Zeroshot CLIP correctly selects the referent object 7 out of 11 times, while LAGOR selects the correct referent 9 out of 11 times. LAGOR corrects CLIP's mistakes on expressions like *Mug with the thin metal handle* where multiple views expose potentially occluded parts like *handle*, as well as when information gathering makes decisions more confident, such as *Orange box* when both objects have orange components, but one object is orange from all views. These experiments introduce a number of new sources of noise not present in the simulated setting, for example changes in camera angle, shifting lighting during rotation, and automatic segmentation to account for cluttered scenes. For example, segmentation leads to the creation of voids inside of objects not present in SNARE. View extraction details, referring expressions, and extracted views are shown in Section 7.6.

## 6 Conclusions

We introduce **S**hape**N**et **A**nnotated with **R**eferring **E**xpressions (SNARE), a challenge task to ground language to 3D object models. We show that fine-tuned, massively pretrained vision and language models fall short of human performance at identifying object referents of natural language expressions by a wide margin, while still achieving well above random chance performance (Table 2). Models improve as more 2D views of 3D objects are available, but a robot can only capture one view at a time. We introduce **La**nguage **G**rounding through **O**bject **R**otation (LAGOR), a model that considers two views of candidate referent objects trained with current view estimation as an auxiliary loss, and builds on a pre-trained CLIP model. We transfer the trained LAGOR model to a physical robot to study its generalization to noisy images of physical objects.

In the future, a rotation *policy* could be learned to examine up to $N$ views across $M$ candidate objects subject to a referring expression, rather than performing a random rotation to a novel view. By assigning rotation actions expected discriminative values using a trained rotation module (though that could introduce a cyclic dependency with the MATCH module), and penalties based on the time it takes a particular hardware to achieve an object rotation, one could learn a POMDP policy aggregating object views seen so far to decide whether to next obtain a new view or make a guess about the referent object. Obtaining different object views could be done either by picking up and rotating the object or by moving the robot camera to see the object from another angle. Further, since 3D meshes are available for each object, it may be possible to extract shape-level information using a PointNet [49], similar to experiments attempted by ShapeGlot [34]. Because we target physical robot applications, we may be able to tie language to sets of *graspable points* encoding gripper orientation and position [44] for each object.

## Acknowledgements

We would like to thank the anonymous CoRL 2021 reviewers for their feedback and suggestions. This paper was made stronger by the review and revise model adopted by CoRL this year via OpenReview. This work was funded in part by ONR under award #1140209-405780.

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
