# OpenReview forum: "Language Grounding with 3D Objects"
_robot-learning.org/CoRL/2021/Conference — CoRL2021 Poster_

### Official Review · Reviewer_ch3k · 2021-07-23

**Originality:** Good
**Technical Quality:** Good
**Clarity Of Presentation:** Good
**Impact:** 3

**Recommendation:**

Weak Accept: I recommend accepting the paper, but will not argue for my recommendation if the majority of other reviewers have a different opinion.

**Summary:**

* Introduces SNARE, a new benchmark for grounding natural language referring expressions to distinguish 3D objects.
* Introduces the concept of 'blindfolded descriptions', a complementary and challenging linguistic space. Introduces a language annotation process that explicitly gathers these (less collected) tactile natural language descriptions of objects.
* Introduce several CLIP-based models for distinguishing objects.

**Issues:**

This paper would benefit from:
* Cleaned up exposition.
* Statistically valid experiments or toned down interpretation of the existing experiments.
* Highlighting the seemingly impressive zero shot performance of CLIP-style models on tactile descriptions.

**Reviewer Expertise:**

Good: General knowledge of the area

**Strengths And Weaknesses:**

[Strengths]
* Good motivation: imbue robots with three dimensional notions of language grounding as opposed to purely visual ones.
* Good description of the CLIP-based models for distinguishing objects and the auxiliary losses. I feel like this carries enough information to begin implementing.
* Informative related work section highlighting how this paper sits at the intersection of multiple active fields.
* Raises an important question: does modern visual language grounding have important limitations for language-conditioned robotic learning, i.e. domain shift between canonical depictions of objects in images and objects as seen through a robot camera?

[Weaknesses]
* LAGOR has quite strict requirements for training (8 simultaneous labeled views of each object at 45 degree offsets), and seems to offer minor benefits in performance over single-view trained methods or pretrained methods. In table 1, we see 79.4% (LAGOR) vs 78.5% (MATCH) averaged over visual and blind descriptions.
* Claims in the conclusion like L303: "We find that LAGOR is able to identify referent objects using only an initial and final view more accurately than models utilizing 360 views" seem quite strong, given the experiments.
* Claims like L297 "We show that fine-tuned, massively pretrained vision and language models fall short of human performance at identifying object referents of natural language expressions by a wide margin (Table 2)." seem odd. You are taking models trained on one domain (massive internet collections) and applying them in zero shot to new domain (simulated object) data. If anything, I'd say the gap between the zero shot performance and the performance of models finetuned on in-domain SNARE data is much smaller than I would have expected.
The more important takeaway from this for me is that pretrained models actually do very well on the considered benchmark. This seems like it could be emphasized more instead of focusing on the relatively minor gains LAGOR offers.


**Summary Of Recommendation:**

This paper has some issues with exposition, but presents interesting results showing how large pretrained language models (and their finetuned variants) handle referent-picking challenges when the language used is more tactile than visual. As pointed out by this work, this different mode of object description may be more likely in the real world than in the large datasets CLIP-style models are trained on. Despite this supposed domain shift, the results show that pretrained models fare surprisingly well in zero shot. This seems like an interesting insight for the broader community.

UPDATE:

* I highly appreciate the additional experiments and improved statistical rigor, as well as the removal of overly strong interpretations.
* In regards to my prior comment "LAGOR has quite strict requirements for training (8 simultaneous labeled views of each object at 45 degree offsets)". I appreciate the clarification that not all need to be present at training time, but computational cost was not what I was concerned with. I think the criticism remains valid that requiring 8 labeled views (even if only two are sampled for each training instance) represents a very significant additional assumption to the overall training process, for pretty minor gains over baselines. In general, I think this limits its applicability to many new scenarios.

---

> ### Author Response · Authors · 2021-08-31
> **Individual response to Reviewer ch3k**
>
> > No confidence intervals are provided over multiple training runs, so it's not easy for the reader to decide if this is a true improvement or just sampling error.
>
> Thanks for pointing this out; we have added a reply to all reviewers that discusses our new, more robust experiments.
>
> > LAGOR has quite strict requirements for training (8 simultaneous labeled views of each object at 45 degree offsets), and seems to offer minor benefits in performance over single-view trained methods or pretrained methods. In table 1, we see 79.4% (LAGOR) vs 78.5% (MATCH) averaged over visual and blind descriptions.
>
> This is a good point, and we’ve made revisions to hopefully show the importance of our results. See the revised numbers in Table 2: while the difference between two-view LaGOR and two-view MATCH is small, it is statistically significant.
>
> While LaGOR uses 8 views of objects from 45 degrees each, it samples two and maxpools those representations from a pre-trained CLIP, and so any two views suffice, not just from those canonical 8, as shown in the generalization experiments to physical objects imaged with a robot’s camera. LaGOR’s view estimation losses do require fixed views across the training objects, but other kinds of 3D information could be used in place of view estimation. For example, future modeling attempts could directly consume a view of the ShapeNet 3D mesh via a PointNet.
>
> What we consider more striking is the gap in performance between off-the-shelf models (e.g. ViLBERT, which is gigantic, and LaGOR, which is based on the smaller CLIP representations with sensible aux losses) trained on multiple views of 3D meshes versus human ability on SNARE.
>
> > Claims in the conclusion like L303: "We find that LAGOR is able to identify referent objects using only an initial and final view more accurately than models utilizing 360 views" seem quite strong, given the experiments.
>
> We agree that these claims were too strong. Especially in light of the new results and our use of statistical significance testing, we have removed and better-scoped these claims as appropriate throughout the paper.
>
> > Claims like L297 "We show that fine-tuned, massively pretrained vision and language models fall short of human performance at identifying object referents of natural language expressions by a wide margin (Table 2)." seem odd. You are taking models trained on one domain (massive internet collections) and applying them in zero shot to new domain (simulated object) data. If anything, I'd say the gap between the zero shot performance and the performance of models finetuned on in-domain SNARE data is much smaller than I would have expected. The more important takeaway from this for me is that pretrained models actually do very well on the considered benchmark. This seems like it could be emphasized more instead of focusing on the relatively minor gains LAGOR offers.
>
> We agree it’s impressive that zero-shot CLIP is able to achieve relatively high performance out of the box. At the same time, given CLIP’s extensive transfer results, it’s not surprising that it can be used as a good initial scoring mechanism for language and vision alignment. We have re-framed the contribution from LaGOR throughout the paper given the updated results in Table 2 and thoughts from the reviewer. We hope that the narrative more clearly centers SNARE as the primary contribution of this paper, with zero shot CLIP, MATCH, and LaGOR more clearly probes into the difficulty of getting human-like performance on SNARE through sensible modeling attempts.

---

### Official Review · Reviewer_9Pxm · 2021-07-23

**Originality:** Good
**Technical Quality:** Good
**Clarity Of Presentation:** Good
**Impact:** 3

**Recommendation:**

Weak Accept: I recommend accepting the paper, but will not argue for my recommendation if the majority of other reviewers have a different opinion.

**Summary:**

The authors collected a dataset (SNARE) with 3D object models and natural language referring expressions. Moreover, they propose the MATCH method to predict the described object among two candidate objects (one target and one distractor). Using this method, they show that using multiple views to select the correct object improves over single-view object selection. They further improve the MATCH model and suggest the LAGOR method estimating the current view of objects to improve the performance.

**Issues:**

- The CLIP baseline method is not fine-tuned in the suggested dataset. Are there any specific reasons to only report the zero-shot performances of this baseline?
- In the instructions of MTUR workers (provided in the appendix), the workers were instructed that visual descriptions can involve the name of the objects, shapes, and colors; by contrast, blindfolded descriptions can not involve colors but can involve shape and parts. However, in the paper manuscript, this is provided as an estimation ("We estimated that blindfolded expressions use more shape words 177 (14% vs 5% of all words) while visual use more color words (22% vs 1% of all words)"). It would be better to acknowledge readers that these were the instructions that should be followed by the MTURK workers.
- It would be nice to include further information about the training of MATCH and LAGOR methods. For instance, what were the negative samples during training? Were they the distractor objects? Did you train these methods for a single view and include the max-pooling during testing for multi-views?
- In Table 2, the results are reported for single, two, and 360. Did you use the 3D model of the objects as an input for the results of 360? Or was it many 2D views of the objects? If it is the second case, the paper's title and abstract might be misleading because the suggested systems are not trained/evaluated on 3D object models but multiple 2D views of the objects.
- In the robotics demonstration results (Section 5.2), did you use a single view both for CLIP and LAGOR? Or was it a single view for CLIP and two views for LAGOR? If the second one is the case, why not provide two views to both of the methods? Further, the accuracy of LAGOR is reported as 50%; isn't it the random chance level (selecting one of two candidates)?
- It would be nice to include a reference to the statement in line 51 ("However, internet images suffer from a type of reporting bias").

-------------------------------------------------------------------------------------------------------------------------------------------------------------------------------------

Post-Rebuttal Comments:

I would like to thank the authors for replying to my comments in detail. The revisions are helpful to highlight the contributions and interpret the results. However, I still have the following concerns after the rebuttal:
- I think simplifying the problem by selecting the target object among two candidates might be the reason for the high accuracies of all the methods shown in Table 2. As also pointed out by Reviewer ch3k, the performances of the zero-shot CLIP models are high on the SNARE benchmark, and this might be because of the high chance level (i.e., 50%). I think the system needs to be tested with more than two candidate objects to present the advantages of estimating the current view of objects and be usable in real-world applications.
- In the robotics demonstration, the authors present the results for a single view for the zero-shot CLIP model and two views for LAGOR. For a fair comparison, it would be better to show these results for the zero-shot CLIP, MATCH, and LAGOR methods with a single view and two views (if resolving the technical issues is possible).

**Reviewer Expertise:**

Good: General knowledge of the area

**Strengths And Weaknesses:**

Strengths:
- The suggested dataset includes both visual descriptions that focus on colors and object categories and blindfolded descriptions that focus on shapes and parts. Computer vision studies commonly suggest visual descriptions, but blindfolded descriptions do not often address existing studies. The blindfolded descriptions suggested in the dataset can be useful when the robot is verbally instructed to grasp a specific object. Moreover, this dataset includes many object categories (262 in total) compared to the previous works.
- The paper presents interesting findings. The results show that the performance of the MATCH method increases when two views of an object are used instead of a single view, but the accuracies do not increase when the 360-degree view is provided. Further, the authors show that estimating the current view helps identifying referred objects.

Weaknesses:
- The suggested system chooses which of two objects is being referenced by a natural language description.  Only having two candidates oversimplifies the problem, given that there might be many distractors in real-world environments. Because of this simplification, one of the critical features of referring expressions (i.e., spatial relations between objects) is excluded in the paper.
- The paper suggests rotating the objects before identifying the described one because some parts of the object might not be visible from the robot's viewpoint, and these invisible features might be used in the user's description. This is a very valid point, but the suggested system rotates the objects even if it is not necessary. In other words, if the visible part is enough to predict the referred object, the system still performs the rotation to make a prediction.

**Summary Of Recommendation:**

Although the paper simplifies the problem and the method might perform unnecessary rotations, it suggests a valuable dataset and presents very interesting findings. These are the main reasons for my recommendation (see the strengths and weaknesses for details).

---

> ### Author Response · Authors · 2021-08-31
> **Individual response to Reviewer 9Pxm (1/n)**
>
> > Only having two candidates oversimplifies the problem, given that there might be many distractors in real-world environments. Because of this simplification, one of the critical features of referring expressions (i.e., spatial relations between objects) is excluded in the paper.
>
> We agree that spatial relations are an important discriminative feature in many domains with referring expressions (e.g., cluttered tabletops). There are dedicated datasets and papers for studying referring expressions, such as CLEVR [https://cs.stanford.edu/people/jcjohns/clevr/] and “Learning Interpretable Spatial Operations in a Rich 3D Blocks World” [https://arxiv.org/abs/1712.03463]. We feel SNARE offers a complementary contribution: language expressions gathered to differentiate between related 3D meshes without the benefit of discriminating spatial relations that can render in-object descriptions less valuable. We would argue that the gap between trained model and human performance (10% across SNARE; nearly 15% on the non-visual subset) on a binary choice task suggests it is not oversimplified.
>
> > The paper suggests rotating the objects before identifying the described one because some parts of the object might not be visible from the robot's viewpoint, and these invisible features might be used in the user's description. This is a very valid point, but the suggested system rotates the objects even if it is not necessary. In other words, if the visible part is enough to predict the referred object, the system still performs the rotation to make a prediction.
>
> We agree that if the system is already confident, it doesn’t make sense for it to bother getting another view. In Section 6, we address the possibility of learning a policy for getting views from different object rotations only as needed, including the possibility of predicting which view would be most helpful to rotate each object to. We consider learning such a policy as future work that could leverage the SNARE annotations and task setup.
>
> > The CLIP baseline method is not fine-tuned in the suggested dataset. Are there any specific reasons to only report the zero-shot performances of this baseline?
>
> We have clarified in Revision Section 4.1 that the MATCH model is, effectively, a fine-tuned CLIP model for the classification task in SNARE. Rather than back-propagating through all of CLIP, we use off-the-shelf CLIP representations as input to learnable layers that fine-tune those representations further for the SNARE task. Typically, full backpropagation through large models is avoided to prevent overfitting of initial layers to the training set. In short: we report both CLIP zero-shot performance and CLIP fine-tuned performance in the form of the MATCH model.
>
> > In the instructions of MTUR workers (provided in the appendix), the workers were instructed that visual descriptions can involve the name of the objects, shapes, and colors; by contrast, blindfolded descriptions can not involve colors but can involve shape and parts. However, in the paper manuscript, this is provided as an estimation ("We estimated that blindfolded expressions use more shape words 177 (14% vs 5% of all words) while visual use more color words (22% vs 1% of all words)"). It would be better to acknowledge readers that these were the instructions that should be followed by the MTURK workers.
>
> This is a great point. We have revised the description of the SNARE data to mention these priming words and their relationship to the final distribution of descriptors explicitly (Revision Section 3).

---

> ### Author Response · Authors · 2021-08-31
> **Individual response to Reviewer 9Pxm (2/n)**
>
> > It would be nice to include further information about the training of MATCH and LAGOR methods. For instance, what were the negative samples during training? Were they the distractor objects? Did you train these methods for a single view and include the max-pooling during testing for multi-views?
>
> MATCH and LaGOR are trained using supervised learning directly on the object prediction task of SNARE. Given two objects (1 positive & 1 negative) and one referring expression, both MATCH and LaGOR are trained with cross entropy loss to predict the referent object. The referring expressions were collected with both the “positive” referent object and “negative” distractor object visible to the annotator to ensure that the expression is able to differentiate the object instances. In other words, these are strong negatives tied to the referring expression for the positive referent.
>
> It would be possible to explore sampling additional negative examples per positive object and associated referring expression, but in these initial experiments we only use the hard negatives elicited in a way that ensures the referring expression refers to the positive and not the negative object unambiguously (See the discussion on quality control in Section 7.1).
>
> Multiple views are pooled (maxpool, in the main results) such that MATCH and LaGOR both take in a single vector representing each object to make a prediction decision.
>
> For multiple view models, for example MATCH with two views or MATCH with all eight views, training and evaluation are both done with that number of views pooled via maxpool. LaGOR always takes in two views and adds an auxiliary prediction of the initial (and, since revision, the final) view among those two representing the object before and after a random rotation --- or, in the case of the physical robot demonstration, an arbitrary additional view via moving the wrist-mounted camera.
>
> > In Table 2, the results are reported for single, two, and 360. Did you use the 3D model of the objects as an input for the results of 360? Or was it many 2D views of the objects? If it is the second case, the paper's title and abstract might be misleading because the suggested systems are not trained/evaluated on 3D object models but multiple 2D views of the objects.
>
> Eight 2D views of the objects are used for the Table 2 results previously labeled “360” views. To mitigate this confusion, we’ve revised Table 2 from “360”->”All” to indicate all 8 fixed views of the object are used representing the rendering of the object at 45 degree angle increments. We have revised the abstract’s discussion of the modeling results to make it clearer that current models are not taking in full 3D meshes but snapshots.
>
> We feel the paper title is appropriate for SNARE, the main contribution, because the rendered objects were shown as rotating 3D models to annotators. Because off-the-shelf methods in language and vision are based on images, not 3D surfaces, our initial models are meant to demonstrate the large gap between such 2D understanding and human-level performance on SNARE. Language grounding with 3D objects, from a modeling perspective, is nascent, and SNARE provides a testbed for future modeling attempts.
>
>
> > In the robotics demonstration results (Section 5.2), did you use a single view both for CLIP and LAGOR? Or was it a single view for CLIP and two views for LAGOR? If the second one is the case, why not provide two views to both of the methods? Further, the accuracy of LAGOR is reported as 50%; isn't it the random chance level (selecting one of two candidates)?
>
> We use a single view for the CLIP zero-shot model and two views for LaGOR. We agree that giving CLIP access to both views is the better comparison. Unfortunately, we were not able to re-run the robot experiments due to technical issues with the Franka robot -- it crashed and would not restart -- but will aim to have them done for the final camera ready. More unfortunately (because it was preventable), not all the images captured by the robot to obtain the results in Section 5.2 were saved to enable two-view CLIP prediction offline. Instead, just one image per object was saved and used for Figure 11 in the appendix.
>
> Regarding LaGOR’s previously poor robot performance: since the original submission, we discovered a bug in the modeling runs for the robot experiments and re-did them with more objects a week or so after the initial submission earlier this summer. CLIP zero shot accuracy is 7/11 in contrast to LaGOR’s 9/11. These newer results and associated figure can be seen in the revised Section 5.2.

---

> ### Author Response · Authors · 2021-08-31
> **Individual response to Reviewer 9Pxm (3/3)**
>
> > It would be nice to include a reference to the statement in line 51 ("However, internet images suffer from a type of reporting bias").
>
> We meant this sentence as a theme for the next two sentences describing the types of reporting bias that occur in internet images. Specifically:
> “However, internet images suffer from a type of reporting bias.
> These images are captured by sighted humans, from human-centric vantage points [1].
> There is a domain shift between such canonical depictions of objects in images and objects as seen through a robot camera [2, 3], as studied in computer vision between such images and those captured by the blind [4].”
>
> [1] Unbiased look at dataset bias. Torralba, Antonio and Efros, Alexei A. Computer Vision and Pattern Recognition (CVPR). 2011.
>
> [2] First-person vision. Kanade, Takeo and Hebert, Martial. Proceedings of the IEEE. Volume 100, number 8, 2442--2453. 2012.
>
> [3] Sim-to-Real Transfer for Vision-and-Language Navigation. Peter Anderson and Ayush Shrivastava and Joanne Truong and Arjun Majumdar and Devi Parikh and Dhruv Batra and Stefan Lee. Conference on Robot Learning (CoRL). 2020.
>
> [4] VizWiz Grand Challenge: Answering Visual Questions From Blind People. Gurari, Danna and Li, Qing and Stangl, Abigale J. and Guo, Anhong and Lin, Chi and Grauman, Kristen and Luo, Jiebo and Bigham, Jeffrey P. Computer Vision and Pattern Recognition (CVPR). 2018.

---

### Official Review · Reviewer_GQzH · 2021-07-24

**Originality:** Good
**Technical Quality:** Good
**Clarity Of Presentation:** Good
**Impact:** 3

**Recommendation:**

Weak Accept: I recommend accepting the paper, but will not argue for my recommendation if the majority of other reviewers have a different opinion.

**Summary:**

This paper addresses the problem of grounding referring expressions for 3D objects. The authors claim that a single 2D image is not enough to identify an object in the 3D world. To formally study the problem, a new dataset, SNARE, based on ShapeNet objects is introduced to test models’ understanding of both visual and non-visual features. In addition, two CLIP-based models, MATCH and LAGOR, are introduced to align images with text descriptions of objects and to leverage an implicit 3D representation of objects to further promote understanding of non-visual cues.


**Issues:**

Besides questions and suggestions from above, I strongly suggest the authors restructure section 4. At this moment, the section contains information about baselines, hypotheses, proposed methods, experimental designs, and experiment results.


**Reviewer Expertise:**

Good: General knowledge of the area

**Strengths And Weaknesses:**

Strengths:
1. The problem this paper studies is very relevant to the robotics community. As the authors mentioned, it’s difficult for robots to collect complete 360 views of real-world objects. Grounding 3D objects in a small number of 2D images from different views has practical value.
2. The data collection process is carefully designed and produces high-quality data. It’s interesting that the authors deliberately elicit specific descriptions of objects by pairing objects. The wordnet analysis of the dataset also provides insights into the semantic content of the language descriptions.
3. The dataset uses the standard ShapeNet objects and the referring expressions also include non-visual features. Therefore the dataset could be useful for future research that incorporates other sensor modalities as well.

Weakness:
1. It’s unclear whether the choice of the pooling method (i.e., max pooling) has affected the performance of models that use images from all viewpoints.
2. The authors should more formally discuss the physical meaning behind the 8 viewpoints. Is there a canonical pose for every category of objects? If not, how can the model learn to predict the correct viewpoint of a novel object, especially since the data are split at the object class level. What is the accuracy of view estimation? Does it correlate to the performance of language grounding?
3. More insights about the design of the LAGOR model is needed to support the authors' claim. It’s unclear why the LAGOR model only predicts the orientation of the first view? Is there a fixed rotational offset between the first and the second views? If not, why shouldn’t the model have access to the pose information for the second view? Will that further help improve the performance of the model?
4. The novelty of the MATCH model is limited. I suspect that the performance difference between the base CLIP and the MATCH model mainly comes from fine tuning, which is expected when using pretrained models.
5. Compared to the around 10% performance increase after fine tuning the CLIP model, the 2% difference between MATCH (two) and LAGOR (two) is relatively small. Due to the small improvement, it’s important that the authors further analyze the cause of it. It would help if the authors could test on different data splits and consistently see this improvement.


**Summary Of Recommendation:**

This paper proposes an interesting solution to a relevant task. However, due to the lack of explanation for the model design and further analysis of the relatively small performance gain, it’s hard to judge the significance of the paper.

---

> ### Author Response · Authors · 2021-08-31
> **Individual response to Reviewer GQzH (1/2)**
>
> > Therefore the dataset could be useful for future research that incorporates other sensor modalities as well.
>
> We’re excited to hear the reviewer feels this way! In fact, we have plans to incorporate simulated grasping data in future modeling attempts on SNARE to try to close the large gap between vision-only models and human performance.
>
> > It’s unclear whether the choice of the pooling method (i.e., max pooling) has affected the performance of models that use images from all viewpoints.
>
> Thanks for pointing this out. We found that mean pooling over viewpoints leads to statistically significantly better validation accuracy for the MATCH models (Revision Section 7.5) than max pooling. While our two-view experiments were re-run with max pooling, this result suggests we can probably squeeze a little more accuracy out of the two-view methods by switching to mean pool. Given the window for rebuttal and time for training and evaluation, though, we aren’t able to get those meanpool + two-view results before the rebuttal deadline. We’ll switch to meanpool for the camera-ready version of the paper. Thanks again for this insight!
>
> > The authors should more formally discuss the physical meaning behind the 8 viewpoints. Is there a canonical pose for every category of objects? If not, how can the model learn to predict the correct viewpoint of a novel object, especially since the data are split at the object class level. What is the accuracy of view estimation? Does it correlate to the performance of language grounding?
>
> The eight viewpoints are 45 degree rotations of ShapeNet object models. There is a canonical “face” view around which these rotations are defined (see Figure 10 for all pre-rendered ShapeNet views; we use just the last 8 of these representing rotation around the object).
>
> The view estimation error (measured as the average distance between the predicted and actual view index in [1, 8]) does not vary much between model retraining runs, and averages 0.635, that is: less than one view index away from the truth on average. This error estimation does not have a meaningful correlation with task accuracy (r=0.166).
>
> > More insights about the design of the LAGOR model is needed to support the authors' claim. It’s unclear why the LAGOR model only predicts the orientation of the first view? Is there a fixed rotational offset between the first and the second views? If not, why shouldn’t the model have access to the pose information for the second view? Will that further help improve the performance of the model?
>
> Thanks for pointing this possibility out. The second viewpoint LaGOR sees is selected randomly, so adding a second viewpoint prediction is sensible. We have revised the paper to reflect that we now predict both the first and second viewpoint as auxiliary losses with LaGOR (Revision Section 4.2). We ablated these aux losses to try predicting only the first view, only the second view, and both views, and found that accuracy did not change more than a tenth of a percent on the validation set, including for visual and non-visual subsets. Thus, either view prediction loss seems sufficient to confer the small benefit we see with LaGOR over the fine-tuned CLIP (MATCH) model by adjusting learned representations to encode some sense of 3D faces.
>
> > The novelty of the MATCH model is limited. I suspect that the performance difference between the base CLIP and the MATCH model mainly comes from fine tuning, which is expected when using pretrained models.
>
> We agree. We do not consider MATCH a novel modeling contribution in this paper, but rather a sensible language-and-vision approach to the SNARE data. We have added a line in Revision Section 4.1 to make explicit that MATCH is a form of fine-tuning on top of CLIP.

---

> > ### Comment · Reviewer_GQzH · 2021-09-04
> > **Post-Rebuttal Comments**
> >
> > I would like to thank the authors for addressing my questions. Considering the clarification of the model design and the additional experiments, I have changed my evaluation.

---

> ### Author Response · Authors · 2021-08-31
> **Individual Response to Reviewer GQzH (2/2)**
>
> > Compared to the around 10% performance increase after fine tuning the CLIP model, the 2% difference between MATCH (two) and LAGOR (two) is relatively small. Due to the small improvement, it’s important that the authors further analyze the cause of it. It would help if the authors could test on different data splits and consistently see this improvement.
>
> We re-ran all our experiments with 10 random seeds to account for some variance. We agree that it would be helpful to run on different data splits, rather than just model training seeds. The way the SNARE data is split (Section 3) adversarially pushes similar object categories into different folds to try to maximize how well we test for generalization. The splitting algorithm is a greedy heuristic that we could add noise to to generate slightly different train/val/test splits, but the existing adversarial split should be close to the “hardest” way of dividing the data. Our goal with SNARE is to emphasize the gap between human performance on this task and the performance of sensible off-the-shelf (VILBERT, CLIP) and trained (MATCH - CLIP fine-tuned for SNARE classification, and LaGOR, CLIP fine-tuned with some reasonable aux losses) models.
>
> > Besides questions and suggestions from above, I strongly suggest the authors restructure section 4. At this moment, the section contains information about baselines, hypotheses, proposed methods, experimental designs, and experiment results.
>
> Thanks for pointing this out. We moved some experimental results discussion that had slipped into Section 4 back into Section 5 where it belongs. In particular, we relocated details about the number of training epochs, MATCH two view versus all view performance, and robot experiment challenges from S4 to S5 in the revision.

---

### Official Review · Reviewer_XGP9 · 2021-07-24

**Originality:** Good
**Technical Quality:** Good
**Clarity Of Presentation:** Good
**Impact:** 2

**Recommendation:**

Weak Reject: I recommend rejecting the paper, but will not argue for my recommendation if the majority of other reviewers have a different opinion.

**Summary:**

Paper presents a novel benchmark, SNARE, which requires the agent to ground a referring expression in natural language to a 3D object. The agent might need to actively rotate the objects to help determine the selection. A CLIP-based baseline model (LAGOR) is proposed. Results demonstrate that LAGOR attains superior performance over other vision-language models (ViLBERT and raw CLIP).

**Issues:**

See [Suggestions & Questions].

**Reviewer Expertise:**

Good: General knowledge of the area

**Strengths And Weaknesses:**

[Strengths]

[+] The main problem (language grounding in 3D) that this new benchmark concerns is fairly interesting. My personal take for the difference between canonical LG and LG in 3D can be primarily about embodiment. The authors did a good job incorporating it in their benchmark and design an interesting task. Although the results are less surprising, it still seems to be a nice contribution to the community.

[+] The paper is overall clear and well-written. The authors make their motivations clear and convincing. The dataset itself and the proposed learner seem ground and I cannot find any significant technical issue.

[Weaknesses]

[-] The author could have made their presentations more clear by including enough details on the embodiment in their main text. After reading this paper, I'm still not quite sure how the four compared learners (ViLBERT, CLIP, MATCH, LAGOR) select the action. The authors are expected to clarify this in a rebuttal.

[-] If my understanding is correct, the only action considered in this benchmark is rotation (along a predefined axis). I won't say it is overclaiming but indeed I find the action space can be rather limited. This can significantly weaken the point of bringing embodiment to language grounding. A hotfix I can think of is to provide the intermediate action sequence for some hard examples (maybe also some failure mode). Otherwise, it can be hard to tell whether the learner masters some embodiment for 3D interaction or just trivialize it with a (hand-crafted?)short-cut strategy (e.g. continuously rotate the object if the uncertainty is high).

[-] The overall performance gap between the baselines and the proposed learner seems to be marginal and the authors do not provide a confidence interval. I strongly encourage the authors to include it in a rebuttal.

[Suggestions & Questions]

The authors are expected to address the issues I mentioned in [Weaknesses].

**Summary Of Recommendation:**

Given the technical quality at this point, I tend to recommend rejection. I may change my mind per other reviewers' opinion and the rebuttal.

---

> ### Author Response · Authors · 2021-08-31
> **Individual Response to Reviewer XGP9**
>
> Thank you for your comments. We hope the improvements made to the paper and the answers to your concerns below have made our contributions clearer.
>
> > The author could have made their presentations more clear by including enough details on the embodiment in their main text. After reading this paper, I'm still not quite sure how the four compared learners (ViLBERT, CLIP, MATCH, LAGOR) select the action. The authors are expected to clarify this in a rebuttal.
>
> We are choosing random rotations for now, although active rotations would be a good thing to look at in future work, as discussed in Conclusions. There are some major challenges in determining how best to rotate the object and what views will be most informative; instead, for now, we are doing viewpoint estimation instead of predicting a relative rotation.
> However, it should be noted that in many real-world robotics contexts (grasping from clutter, kitting, retrieving objects from cubbies or cabinets) we would have very limited ability to control which views the robot has access to. For this reason, we feel it’s valuable to look at the ability to aggregate over random (or arbitrary) viewpoints.
>
> > If my understanding is correct, the only action considered in this benchmark is rotation (along a predefined axis). I won't say it is overclaiming but indeed I find the action space can be rather limited. This can significantly weaken the point of bringing embodiment to language grounding. A hotfix I can think of is to provide the intermediate action sequence for some hard examples (maybe also some failure mode). Otherwise, it can be hard to tell whether the learner masters some embodiment for 3D interaction or just trivialize it with a (hand-crafted?)short-cut strategy (e.g. continuously rotate the object if the uncertainty is high).
>
> We agree that the action space is certainly quite limited. Our goal in this work was not to learn a policy, but to learn best how to use the limited information we are given. Therefore, the system cannot take advantage of a trivial action (“just rotate continuously”). For instance, consider the case of grasping an object in clutter, where the robot might not be able to see the object except from a couple different arbitrary angles. These cases are more common in realistic environments than they are in laboratory settings.
>
> > The overall performance gap between the baselines and the proposed learner seems to be marginal and the authors do not provide a confidence interval. I strongly encourage the authors to include it in a rebuttal.
>
> Thanks for the comment. We’ve addressed this by re-running our experiments with 10 different seeds. Our results are now in Table 2 of the paper. We summarize these results in the response to all reviewers, and have updated the paper discussion to reflect these more robust experiments.

---

### Author Response · Authors · 2021-08-31
**To all: Big thank you + Robust experimental results**

Thank you to the reviewers for their useful comments. We have carefully read all the comments and have revised the paper accordingly. Our primary motivation is the introduction of SNARE as a new benchmark and resource for better understanding the intersection of language, robotics, and 3D understanding. The models are introduced to probe distinctions between visual and non-visual language and show the limitations of current (high powered) visual representations.  Your comments were immensely helpful in clarifying this contribution.

We have also responded individually to each reviewer’s other questions and technical concerns, but wanted to take this space to address the question of model performance differences.

We appreciate the reviewers requesting error bars/multiple runs for models given the percent-splitting differences between some accuracies. In the process of re-running models ten times with different seeds, we discovered a bug introduced by Pytorch-Lightning’s “sanity check” run that fires before training begins, which had poisoned some of the original results in the paper. We want to offer a genuine thank you to each reviewer for variously asking for more experiments and error bars, because we were able to identify and resolve this error and ensure the robustness of results in the paper.

We re-ran all models 10 times with random seeds to estimate variance from initialization and sampling during training. We updated the LaGOR model to perform auxiliary view estimation on both the first and second random view of each object during the SNARE prediction task.

Please revisit Table 2 in the paper for final performance numbers of the zero shot CLIP, MATCH, and LaGOR models. We have updated all sections of the paper discussing results to reflect these new, accurate results. We summarize the two main takeaways results here, paraphrasing the revised Section 5.1:

First, LaGOR statistically significantly outperforms two-view MATCH on both the validation (p=0.0008) and test (p=0.0055) sets.

Second, LaGOR averages higher accuracy than the MATCH that uses all views on every metric; however, these differences are significant for neither the validation (p=0.0844) nor test (p=0.0315) set.

To compare accuracies, we performed an unpaired, Welch's two-tailed t-test on overall accuracy considering both visual and non-visual subsets. We perform five such tests in all: for validation and test, LaGOR against two-view MATCH and LaGOR against all-view MATCH, as well as one additional pooling test. We perform a conservative Bonferroni multiple-comparison correction applied to threshold of p<0.05, which is just to say significance is reached only for tests with p < 0.01, (0.05 threshold / 5 tests).

We sincerely hope that these more robust and statistically backed results assuage each reviewer’s concerns about the previous presentation, and either way, again, extend genuine thanks for requesting them, as they have both strengthened the paper and the experimental results reporting code itself.

We also wanted to make clear we will make SNARE and the modeling code for reproducing all results in the paper publicly available. In the current paper revision we have not included the link to the code repository in order to preserve anonymity.

---

### Meta-Review · Area_Chair_5HUa · 2021-08-03

**Recommendation:** Accept (Poster)
**Confidence:** 4

**Metareview:**

This paper studies an interesting problem, and puts together an interesting dataset that I expect will be impactful in future works. With only a few exceptions, the paper is also well-written and well-motivated. The only exceptions are that the paper includes some claims that are not fully supported (see reviewers XGP9, ch3k, GQzH), and a few details mentioned by reviewers XGP9 and 9Pxm. The paper also proposes several CLIP-based models, and the results of these models are not entirely convincing, especially given the lack of error bars.

Overall, the paper is borderline. I think that the paper currently leans towards accept, but this may change during the author response and reviewer discussion.

---------------------

After the author response period, some of the reviewer's concerns have been addressed, especially the concerns about the robustness of the results. Beyond the comments above, the paper also includes experiments on a real robot platform, which I expect to be appreciated by the CoRL community. I recommend accept.

---

> ### Author Response · Authors · 2021-08-31
> **Added robust experiments and re-scoped claims given those results**
>
> Thanks for your honest analysis of the initial reviews. We hope that the robust experiments presented in the revised version of the paper assuage reviewer concerns, and we are grateful to the AC and reviewers for pushing for those experiments. We believe the revised paper is much stronger, not only for having more carefully supported claims, but for clarity and contribution scope as a result of reviewer feedback and questions.

---

### Decision · Program_Chairs · 2021-09-13

**Decision:**

Accept (Poster)

**Comment:**

This paper studies an interesting problem, and puts together an interesting dataset that I expect will be impactful in future works. With only a few exceptions, the paper is also well-written and well-motivated. The only exceptions are that the paper includes some claims that are not fully supported (see reviewers XGP9, ch3k, GQzH), and a few details mentioned by reviewers XGP9 and 9Pxm. The paper also proposes several CLIP-based models, and the results of these models are not entirely convincing, especially given the lack of error bars.

Overall, the paper is borderline. I think that the paper currently leans towards accept, but this may change during the author response and reviewer discussion.

---------------------

After the author response period, some of the reviewer's concerns have been addressed, especially the concerns about the robustness of the results. Beyond the comments above, the paper also includes experiments on a real robot platform, which I expect to be appreciated by the CoRL community. I recommend accept.